# How do symptoms of each joint contribute to global pain, disease activity and functional disability in rheumatoid arthritis?—A comprehensive association study using a large cohort

Akio Umemoto[1], Hiromu Ito[1,2,3]*, Masayuki Azukizawa[4], Koichi Murata[1,2], Masao Tanaka[2], Takayuki Fujii[1,2], Akira Onishi[2], Hideo Onizawa[2], Shinichiro Ishie[1], Akinori Murakami[1], Kohei Nishitani[1], Kosaku Murakami[5], Hiroyuki Yoshitomi[6], Motomu Hashimoto[2,7], Akio Morinobu[5], Shuichi Matsuda[1]

1 Department of Orthopaedic Surgery, Kyoto University Graduate School of Medicine, Kyoto, Japan, 2 Department of Advanced Medicine for Rheumatic Diseases, Kyoto University Graduate School of Medicine, Kyoto, Japan, 3 Department of Orthopaedic Surgery, Kurashiki Central Hospital, Kurashiki, Japan, 4 Department of Orthopaedic Surgery, National Hospital Organization Himeji Medical Center, Himeji, Japan, 5 Department of Rheumatology and Clinical Immunology, Kyoto University Graduate School of Medicine, Kyoto, Japan, 6 Department of Immunology, Kyoto University Graduate School of Medicine, Kyoto, Japan, 7 Department of Clinical Immunology, Graduate School of Medicine, Osaka Metropolitan University, Osaka, Japan

* hiromu@kuhp.kyoto-u.ac.jp

## Abstract

### Background

Established assessment tools for patients with rheumatoid arthritis (RA), including disease activity scores (DASs), disease activity indexes (DAIs), visual analog scales (VASs), and health assessment questionnaires (HAQs), are widely used. However, comparative associations between joint involvement and disease status assessment tools have rarely been investigated.

### Methods

We included a dataset of 4016 patients from a large RA cohort from 2012 to 2019. The tenderness and swelling of each joint were counted as a symptom, with 70 and 68 affected joints throughout the body, respectively. The relative contribution of various joints to the disease status assessment tools, VAS scores, and functional disability indexes was analyzed using multiple regression analysis.

### Results

The wrist showed the most significant contribution overall, especially in DASs and VASs, while the metacarpophalangeal and proximal interphalangeal joints made significant contributions to DASs and DAIs, but not to VASs and HAQs. The shoulder and the elbow

**Data Availability Statement:** All relevant data are within the paper and its Supporting Information files.

**Funding:** The authors received no specific funding for this work.

**Competing interests:** I have read the journal's policy and the authors of this manuscript have the following competing interests: AO, HO, TF, KMurat, and MT belong to the department financially supported by two local governments in Japan (Nagahama City, Shiga and Toyooka City, Hyogo) and five pharmaceutical companies (Mitsubishi Tanabe Pharma Co., Chugai Pharmaceutical Co., Ltd., AYUMI Pharmaceutical Co., Asahi Kasei Pharma Co. and UCB Japan Co., Ltd.). KURAMA cohort study is supported by grant from Daiichi Sankyo Co. Ltd. HI has received a research grant and/or speaker fee from Bristol-Myers Squibb. MT has received research grants and/or speaker fees from Abbvie Inc., Asahi Kasei Pharma Co., Astellas Pharma Inc., Chugai Pharmaceutical Co. Ltd., Daiichi Sankyo Co., Ltd., Eisai Co. Ltd., Eli Lilly Japan K.K., Janssen Pharmaceutical K.K., Kyowa Kirin Co. Ltd., Pfizer Inc., Taisho Pharmaceutical Co. Ltd., Mitsubishi Tanabe Pharma Co., Teijin Pharma Ltd.,and UCB Japan Co. Ltd. AO reports grants from Pfizer Inc., Bristol-Myers Squibb., Ayumi, The Health Care Science Institute, and Advantest, and personal fees from Asahi Kasei Pharma Co., Chugai Pharmaceutical Co. Ltd., Eli Lilly Japan K.K, Ono Pharmaceutical Co., Mitsubishi Tanabe Pharma, Eisai Co. Ltd., Abbvie Inc., Takeda Pharmaceutical Company Limited, and Daiichi Sankyo Co. Ltd.. MH receives grants and /or speaker fees from Abbvie Inc., Asahi Kasei Pharma Co., Astellas Pharma Inc., AYUMI Pharmaceutical Co., Bristol-Meyers, Chugai Pharmaceutical Co. Ltd, Daiichi Sankyo Co. Ltd, EA Pharma, Eisai Co. Ltd., Eli Lilly Japan K.K., Nihon Shinyaku, Novartis Pharma, and Mitsubishi Tanabe Pharma Co.. AU, MA, SI, AMu, KN, KMurak, HY, AMo and SM declared no conflicts of interest. This study is conducted as an investigator initiate study. The sponsors were not involved in the study design; in the collection, analysis, interpretation of data; in the writing of this manuscript; or in the decision to submit the article for publication. The authors, their immediate families, and any research foundations with which they are affiliated have not received any financial payments or other benefits from any commercial entity related to the subject of this article.

significantly contributed to HAQs, but only the shoulder did to the VASs. The knee universally contributed to all of the tools, but the ankle played a minor but important role in most assessment tools, especially in HAQs. Similar but different contribution ratios were found between the sets of DASs, DAIs, VASs, or HAQs.

## Conclusions

Each joint makes a unique contribution to these assessment tools. The improvement or aggravation of symptoms in each joint affects the assessment tools in different manners.

## Background

Rheumatoid arthritis (RA) is a chronic inflammatory disease that invariably causes swelling and pain in multiple joints, leading to joint destruction and deformity, resulting in impaired daily activities [1]. The joints affected in RA are mainly small joints, according to the European League against Rheumatism (EULAR)/American College of Rheumatology (ACR) classification criteria for RA [2], but larger joints are not excluded. Several studies have reported the prevalence rates of affected joints throughout the body, including the small joints such as the metacarpophalangeal (MCP) and proximal interphalangeal (PIP) joints. Additionally, the large joints such as the knee, shoulder, elbow, or even ankle joints are also often affected [3–5]. Therefore, treat-to-target strategy aims to evaluate disease status to make decisions regarding the treatment, and the disease status assessment tools take joints throughout the body into consideration with scores of 28, 44, or 58 joints in Disease Activity Score (DAS), ACR core set, Simplified Disease Activity Index (SDAI), and Clinical Disease Activity Index (CDAI).

Disease activities, using tools such as DAS in 28 joints (DAS28) and SDAI, were calculated from the "patient global" visual analog scale (PG-VAS), a biomarker of C-reactive protein (CRP) or erythrocyte sedimentation rate (ESR), and the number of joints with tenderness and swelling at the time of evaluation. The values calculated from these have been repeatedly proven to be reliable indicators for assessing the current status of disease activity and predictors for the development of joint destruction [6]. However, these disease state assessment tools are not weighted by each joint. The joints evaluated are different in terms of anatomical size, affected prevalence, and possible degrees of joint destruction. Even though the contribution of each component can be mathematically calculated [7], the weighted contributions of individual joints would be totally different in the clinical settings.

Moreover, it seems apparent that affected joints have different contributions to functional disability; a symptomatic knee joint theoretically influences daily activities more than a symptomatic PIP joint [8, 9]. This may also hold true for pain [10], the most typical symptom in patients with RA, and rheumatologists should give careful consideration to this when applying any treatment strategy to deal with the patients and their symptoms, but few studies have focused on this topic thus far.

A few previous studies have reported the association between symptoms of specific joints and the variables used for the evaluation of "systemic" disease activity or functional disability. A previous study that used the ACR Core dataset reported that the shoulder, knee, and elbow joints were the greatest contributors to the visual analog scale (VAS) score for pain (67.8%), PG-VAS (67.3%), and health assessment questionnaire (HAQ) scores (71.9%) [10]. Another retrospective study with a 3-year follow-up showed the effect of individual joint impairment on functional capacity; worsening in the Japanese version of the HAQ scores was most

prominently observed in the wrists (31%), shoulders (21%), knees (13%), and ankles (10%) [11]. Another study reported the association between bilateral and unilateral joint disease and modified HAQ (mHAQ) scores and the differences in the joints affected; the mHAQ score was significantly associated with shoulder, elbow, wrist, knee, and ankle joint disease [12]. However, comparative associations between joint involvement and assessment tools for disease activity with VASs and HAQs have rarely been investigated.

To clarify the contributions of each specific joint to these clinical assessment tools, we hypothesized that symptoms of each joint contribute to these assessment tools in different manners. To prove this, we used a large RA cohort with precise data and set the objectives of this study as follows: (1) to analyze the contribution of each joint to the assessment tools of disease activity, VAS, and functional disability, (2) to compare the contribution of each joint to each assessment tool, and (3) to visualize the overall association of symptomatic joints and assessment tools.

## Methods

### Patients and methods

We included 4016 patients from the Kyoto University Rheumatoid Arthritis Management Alliance (KURAMA) cohort from 2012 to 2019. The purpose of the cohort study was to manage RA and to apply the clinical and laboratory data obtained in the cohort to clinical research. Patients' clinical and biological data were recorded prospectively at baseline and at each follow-up visit [13, 14]. This study was designed in accordance with the Declaration of Helsinki and approved by the Ethics Committee of the Graduate School of Medicine, Kyoto University (E1308, R0357). All participants provided written informed consent for enrollment in the study.

### Clinical and laboratory evaluation

Patient data were cross-sectionally recorded between 2012 and 2019. Age, symptom duration, Steinbrocker radiological stage, ACR class, DAS based on CRP (DAS28-CRP), disease activity score based on ESR (DAS28-ESR), the values of anti-cyclic citrullinated peptide (anti-CCP) antibody, rheumatoid factor (RF), CRP, and ESR, PG-VAS, pain VAS, HAQ-disability index (HAQ-DI), mHAQ, CDAI, SDAI, tender joint count (TJC), and swollen joint count (SJC). Autoantibody titers of 4.5 U/mL or higher for anti-CCP antibody and 15 IU/mL or higher for RF were considered positive. Tenderness and swelling were comprehensively recorded in 70 joints for TJC and 68 joints for SJC throughout the body, including the joints evaluated by the ACR Core Data Set and the distal interphalangeal (DIP) joints of the fingers. Information on the drugs used by each patient was also obtained.

### Statistical analysis

Patient characteristics are expressed as mean ± standard deviation for continuous variables and percentages (%) for categorical variables. Differences between the 2013–2019 and 2012 KURAMA cohorts were analyzed using Pearson's chi-square test (categorical variables) or Student's t-test (continuous variables). The 70 joints for tenderness and 68 joints for swelling investigated in this study were classified into 15 joint regions as follows: jaw, shoulder, acromioclavicular joint, sternoclavicular joint, elbow, wrist, carpometacarpal joint, MCP joint, PIP joint, DIP joint, hip, knee, ankle, midfoot, and forefoot. The midfoot joint area included the transverse tarsal joint (Chopart's joint) and tarsometatarsal joint (Lisfranc joint). The forefoot included the metatarsophalangeal (MTP) joints and PIP joints. The scores for each joint were

calculated as follows: 2 points (pain or swelling in bilateral joints), 1 point (pain or swelling in a unilateral joint), and 0 points (no pain or swelling in the joints). The scores of all 15 joint categories were calculated as the sum of all the joints of the left and right extremities.

The relative contributions of various joints to disease activity, such as DAS28-CRP and DAS28-ESR (DAS group), SDAI and CDAI (DAI group), PG-VAS and pain VAS (VAS group), and HAQ-DI and mHAQ (HAQ group), were analyzed by multiple regression analysis. First, all joints were analyzed, and the same analysis was performed using a model in which variables with negative standardized coefficients were excluded. The ratio of partial R-squared values was used to calculate the relative contributions to each joint. Further, Circos Table Viewer v. 0.63–9 was used to visualize the relationship between joint symptoms and assessment indexes [15]. Statistical analysis was performed using JMP Pro software, version 14 (SAS Institute, Cary, NC, USA) for the main analyses, and IBM SPSS Statistics version 24 (IBM Corp., Armonk, NY, USA) to verify the results obtained using the JMP Pro software.

First, we analyzed the dataset from 2013 to 2019 as the main study. Then, we performed the same analysis using the 2012 dataset as a validation study and compared the results with the main results (S1 Fig).

## Results

### Patient characteristics

Table 1 shows the clinical characteristics of the study population in the main study (2013–2019) and those of the validation study. In the main study, the mean patient age was 63.7 ± 12.7 years, mean symptom duration was 14.3 ± 15.1 years, mean DAS28-CRP score was 2.06 ± 0.89, and percentage of women was 83.3%. Most patients had long-standing RA with radiographically advanced disease; however, disease activity was well controlled with a treat-to-target strategy using methotrexate (MTX; used by 68.3% of the patients) and/or biological disease-modifying antirheumatic drugs (bDMARDs) or targeted synthetic DMARDs (tsDMARDs; used by 66.0% of the patients). We observed slight differences between men and women in age, symptom duration (years), DAS28-ESR, and CRP but not in anti-CCP antibody and RF positivity, DAS 28-CRP, and PG-VAS (data not shown). On the other hand, the study subjects in the 2012 cohort had worse disease activity and functional disease activity. (Table 1).

### Differential contributions of each joint to each assessment index

First, we analyzed the contribution of joint symptoms using a multivariable analysis. Table 2–1 (DAS28-CRP) shows that the shoulder, elbow, wrist, MCP, PIP, and knee joints contributed more than the other joints, even when joints with negative β were excluded. Contrastingly, Table 2–2 and 2-3 (PG-VAS and HAQ-DI, respectively) show that larger joints such as the shoulder, elbow, wrist, knee, and ankle joints contributed more than smaller joints such as the MCP and PIP joints. We also found that the β and t values of the same joint were different between DAS 28-CRP, PG-VAS, and HAQ-DI, indicating the differential contribution of each joint to each assessment index. Subsequently, we analyzed all the assessment indexes and compared them with each other using a partial R square. S1-1 to S1-5 Tables in S1 Table show the relative contribution of each joint symptom to each assessment index, and Table 3 and S2–S5 Figs summarize the relative contribution of each joint symptom, excluding joints with values less than 0.01, in all assessment indexes. Joint symptoms contributed more to the assessment indexes (nearly 70% of the indexes) in the DAS and DAI groups but less in the VAS and HAQ groups (nearly 15%) (S2 Fig), indicating the weighted significance of joint symptoms on these assessment indexes and possible future development of joint destruction.

**Table 1. Clinical characteristics of the study population in the 2013–2019 and 2012 KURAMA cohort.**

|  | 2013–2019 | 2012 | P-Value |
|---|---|---|---|
|  | n = 3485 | n = 370 |  |
| Age, years | 63.7 ± 12.7 | 62.9 ± 13.0 | 0.273 |
| Symptom duration, years | 14.6 ± 15.1 | 14.9 ± 11.5 | 0.592 |
| Stage (I /II /III /IV), % | 24.9/ 26.6/ 16.7/ 31.7 | 16.0/ 23.8/ 17.1/ 43.1 | < 0.0001 |
| Class (I /II /III /IV), % | 40.2/ 44.8/ 9.2/ 0.3 | 24.3/ 57.0/ 17.8/ 0.8 | < 0.0001 |
| DAS28-CRP | 2.06 ± 0.89 | 2.53 ± 1.04 | < 0.0001 |
| DAS28-ESR | 2.78 ± 1.10 | 3.21 ± 1.16 | < 0.0001 |
| Anti-CCP antibody, U/mL | 217.0 ± 407.3 | 123.8 ± 113.5 | < 0.0001 |
| The rate of anti-CCP antibody positive, % | 77.2 | 82.3 | 0.0248 |
| RF, mg/dL | 111.7 ± 294.2 | 87.9 ± 145.6 | 0.0093 |
| The rate of RF positive, % | 76.4 | 76.0 | 0.0002 |
| CRP, mg/L | 0.42 ± 1.03 | 0.59 ± 1.06 | 0.0033 |
| ESR 1h, mm/h | 22.9 ± 19.4 | 27.0 ± 22.4 | 0.0009 |
| PG-VAS, 0-100mm | 29.0 ± 24.8 | 34.7 ± 25.7 | < 0.0001 |
| Pain VAS, 0-100mm | 25.5 ± 25.2 | 32.2 ± 27.0 | < 0.0001 |
| HAQ-DI | 0.65 ± 0.73 | 0.88 ± 0.81 | < 0.0001 |
| mHAQ | 0.36 ± 0.54 | 0.51 ± 0.59 | < 0.0001 |
| CDAI | 5.42 ± 5.48 | 7.78 ± 6.20 | < 0.0001 |
| SDAI | 5.87 ± 5.89 | 8.41 ± 6.72 | < 0.0001 |
| TJC | 0.94 ± 1.97 | 1.23 ± 1.88 | 0.0052 |
| SJC | 0.85 ± 1.68 | 1.39 ± 2.07 | < 0.0001 |
| The use of PSL, % | 26.5 | 40.5 | < 0.0001 |
| PSL dose, mg/day | 1.32 ± 3.69 | 1.94 ± 3.00 | 0.0002 |
| The use of MTX, % | 68.3 | 70.8 | 0.3160 |
| MTX dose, mg/week | 5.27 ± 4.21 | 5.10 ± 4.23 | 0.4809 |
| The use of bDMARD or tsDMARD, % | 66.0 | 29.5 | < 0.0001 |

Then, we focused on the overall contribution of joint symptoms alone and calculated and analyzed the ratio of partial R-squared for each joint, excluding the residuals (Fig 1). Disease activity indexes (DAS and DAI groups) showed a higher contribution of the wrist; moderate contribution of the shoulder, elbow, MCP joint, and knee; and lower contribution of the PIP joint and ankle. VAS group (PG-VAS and pain VAS) showed the highest contribution of the wrist, second highest of the shoulder, third highest of the knee, moderate contribution of the elbow and the ankle, and much lower contribution of the MCP and the PIP joints. The HAQ group (HAQ-DI and mHAQ) showed a higher contribution of the shoulder, wrist, and elbow; moderate contribution of the knee and ankle; and much lower contrast of the MCP and PIP joints.

## Differences in similar assessment indexes in terms of the contribution of joint symptoms

Next, we compared the relative contributions of the similar assessment tools. DAS28-ESR showed a relatively less contribution of joint symptoms than the other disease activity indexes, such as DAS-CRP, SDAI, and CDAI (S2 Fig). Fig 1 shows that the contribution of the MCP joint was higher in the DAI group than that in the DAS group. In the VAS group, PG-VAS and pain VAS showed similar contributions of the joint symptoms, but the shoulder contributed more to pain VAS whereas the elbow contributed more to PG-VAS, indicating similar

**Table 2. Multivariable association for the contribution of various joint to DAS28-CRP in the 2013–2019 KURAMA cohort. 1. 2.** Multivariable association for the contribution of various joint to PG-VAS in the 2013–2019 KURAMA cohort. **3.** Multivariable association for the contribution of various joint to HAQ-DI in the 2013–2019 KURAMA cohort.

| n = 3485 | All variables | | | | | Exclude variables with negative β | | | | | | |
|---|---|---|---|---|---|---|---|---|---|---|---|---|
| | B | Std. Error | β | t-value | P-value | B | Std. Error | β | t-value | P-value | Partial R-square | %contribution |
| **1.** | | | | | | | | | | | | |
| Jaw | -0.172 | 0.214 | -0.008 | -0.804 | 0.421 | - | - | - | - | - | - | - |
| Shoulder | 0.508 | 0.025 | 0.224 | 20.665 | 0.000 | 0.507 | 0.024 | 0.223 | 20.727 | 0.000 | 0.090 | 13.8 |
| Acromioclavicular | -0.048 | 0.063 | -0.008 | -0.757 | 0.449 | - | - | - | - | - | - | - |
| Sternoclavicular | 0.067 | 0.132 | 0.005 | 0.510 | 0.610 | 0.042 | 0.128 | 0.003 | 0.325 | 0.745 | 0.000 | 0.0 |
| Elbow | 0.413 | 0.021 | 0.212 | 19.289 | 0.000 | 0.410 | 0.021 | 0.211 | 19.366 | 0.000 | 0.093 | 14.2 |
| Wrist | 0.462 | 0.015 | 0.332 | 30.268 | 0.000 | 0.462 | 0.015 | 0.331 | 30.259 | 0.000 | 0.182 | 27.7 |
| CM | 0.098 | 0.044 | 0.023 | 2.230 | 0.026 | 0.094 | 0.044 | 0.022 | 2.138 | 0.033 | 0.004 | 0.5 |
| MCP | 0.322 | 0.017 | 0.208 | 18.652 | 0.000 | 0.323 | 0.017 | 0.208 | 18.732 | 0.000 | 0.096 | 14.5 |
| (P)IP | 0.456 | 0.022 | 0.225 | 21.147 | 0.000 | 0.455 | 0.022 | 0.225 | 21.165 | 0.000 | 0.084 | 12.8 |
| DIP | 0.053 | 0.066 | 0.008 | 0.793 | 0.428 | 0.052 | 0.066 | 0.008 | 0.792 | 0.428 | 0.001 | 0.1 |
| Hip | 0.044 | 0.065 | 0.007 | 0.684 | 0.494 | 0.046 | 0.065 | 0.007 | 0.719 | 0.472 | 0.001 | 0.1 |
| Knee | 0.479 | 0.021 | 0.251 | 22.939 | 0.000 | 0.473 | 0.020 | 0.248 | 23.503 | 0.000 | 0.100 | 15.3 |
| Ankle | 0.042 | 0.021 | 0.022 | 2.004 | 0.045 | 0.039 | 0.021 | 0.020 | 1.876 | 0.061 | 0.005 | 0.8 |
| Midfoot | -0.030 | 0.036 | -0.009 | -0.821 | 0.412 | - | - | - | - | - | - | - |
| Forefoot | 0.017 | 0.015 | 0.011 | 1.087 | 0.277 | 0.016 | 0.015 | 0.011 | 1.010 | 0.313 | 0.002 | 0.2 |
| **2.** | | | | | | | | | | | | |
| Jaw | 0.937 | 9.266 | 0.002 | 0.101 | 0.919 | - | - | - | - | - | - | - |
| Shoulder | 8.765 | 1.047 | 0.137 | 8.370 | 0.000 | 8.751 | 1.047 | 0.137 | 8.358 | 0.000 | 0.033 | 18.3 |
| Acromioclavicular | -5.349 | 2.666 | -0.033 | -2.006 | 0.045 | - | - | - | - | - | - | - |
| Sternoclavicular | 4.018 | 5.678 | 0.011 | 0.708 | 0.479 | 1.591 | 5.548 | 0.004 | 0.287 | 0.774 | 0.000 | 0.1 |
| Elbow | 5.097 | 0.902 | 0.094 | 5.652 | 0.000 | 4.867 | 0.893 | 0.090 | 5.448 | 0.000 | 0.020 | 11.0 |
| Wrist | 7.295 | 0.647 | 0.187 | 11.272 | 0.000 | 7.268 | 0.647 | 0.186 | 11.231 | 0.000 | 0.054 | 30.4 |
| CM | 2.773 | 1.885 | 0.023 | 1.471 | 0.141 | 2.647 | 1.884 | 0.022 | 1.405 | 0.160 | 0.002 | 1.2 |
| MCP | 2.134 | 0.733 | 0.049 | 2.912 | 0.004 | 2.212 | 0.732 | 0.051 | 3.022 | 0.003 | 0.010 | 5.4 |
| (P)IP | 2.571 | 0.912 | 0.045 | 2.819 | 0.005 | 2.457 | 0.910 | 0.043 | 2.698 | 0.007 | 0.005 | 2.9 |
| DIP | 0.697 | 2.840 | 0.004 | 0.245 | 0.806 | 0.704 | 2.841 | 0.004 | 0.248 | 0.804 | 0.000 | 0.1 |
| Hip | 7.422 | 2.675 | 0.043 | 2.775 | 0.006 | 7.367 | 2.675 | 0.043 | 2.754 | 0.006 | 0.004 | 2.1 |
| Knee | 7.001 | 0.879 | 0.132 | 7.960 | 0.000 | 6.945 | 0.878 | 0.131 | 7.911 | 0.000 | 0.030 | 16.5 |
| Ankle | 3.577 | 0.895 | 0.066 | 3.995 | 0.000 | 3.393 | 0.890 | 0.062 | 3.812 | 0.000 | 0.012 | 6.6 |
| Midfoot | 1.834 | 1.522 | 0.020 | 1.205 | 0.228 | 1.918 | 1.522 | 0.021 | 1.261 | 0.207 | 0.003 | 1.5 |
| Forefoot | 2.434 | 0.661 | 0.059 | 3.685 | 0.000 | 2.387 | 0.660 | 0.057 | 3.613 | 0.000 | 0.007 | 4.1 |
| **3.** | | | | | | | | | | | | |
| Jaw | 0.075 | 0.271 | 0.004 | 0.275 | 0.783 | 0.071 | 0.271 | 0.004 | 0.261 | 0.794 | 0.000 | 0.1 |
| Shoulder | 0.327 | 0.031 | 0.172 | 10.676 | 0.000 | 0.326 | 0.031 | 0.172 | 10.631 | 0.000 | 0.049 | 24.5 |
| Acromioclavicular | -0.063 | 0.078 | -0.013 | -0.802 | 0.423 | - | - | - | - | - | - | - |
| Sternoclavicular | 0.115 | 0.166 | 0.011 | 0.693 | 0.488 | 0.094 | 0.162 | 0.009 | 0.578 | 0.563 | 0.001 | 0.3 |
| Elbow | 0.228 | 0.026 | 0.142 | 8.657 | 0.000 | 0.221 | 0.026 | 0.138 | 8.443 | 0.000 | 0.036 | 18.4 |
| Wrist | 0.198 | 0.019 | 0.171 | 10.444 | 0.000 | 0.197 | 0.019 | 0.170 | 10.394 | 0.000 | 0.049 | 24.6 |
| CM | 0.086 | 0.055 | 0.024 | 1.562 | 0.118 | 0.079 | 0.055 | 0.022 | 1.427 | 0.154 | 0.002 | 1.1 |
| MCP | 0.056 | 0.021 | 0.043 | 2.590 | 0.010 | 0.044 | 0.021 | 0.034 | 2.095 | 0.036 | 0.006 | 2.9 |
| (P)IP | -0.068 | 0.027 | -0.040 | -2.529 | 0.011 | - | - | - | - | - | - | - |
| DIP | -0.162 | 0.083 | -0.030 | -1.948 | 0.051 | - | - | - | - | - | - | - |
| Hip | 0.196 | 0.078 | 0.038 | 2.500 | 0.012 | 0.192 | 0.078 | 0.038 | 2.454 | 0.014 | 0.003 | 1.6 |
| Knee | 0.179 | 0.026 | 0.114 | 6.955 | 0.000 | 0.177 | 0.026 | 0.113 | 6.872 | 0.000 | 0.024 | 12.2 |
| Ankle | 0.166 | 0.026 | 0.103 | 6.317 | 0.000 | 0.161 | 0.026 | 0.100 | 6.160 | 0.000 | 0.023 | 11.6 |
| Midfoot | 0.066 | 0.045 | 0.024 | 1.483 | 0.138 | 0.069 | 0.045 | 0.025 | 1.548 | 0.122 | 0.003 | 1.7 |
| Forefoot | 0.037 | 0.019 | 0.030 | 1.909 | 0.056 | 0.029 | 0.019 | 0.024 | 1.528 | 0.127 | 0.002 | 1.2 |

**Table 3. Relative contribution to various joints in the 2013–2019 KURAMA cohort excluding those with negative β.**

|  | DAS28-CRP | DAS28-ESR | CDAI | SDAI | PG-VAS | pain VAS | HAQ-DI | mHAQ |
|---|---|---|---|---|---|---|---|---|
| Jaw | - | - | 0.0 | - | - | 0.3 | 0.1 | 0.0 |
| Shoulder | 13.8 | 14.3 | 11.0 | 11.6 | 18.3 | 21.4 | 24.5 | 27.9 |
| Acromioclavicular | - | 0.0 | 0.1 | - | - | - | - | - |
| Sternoclavicular | 0.0 | 0.0 | 0.2 | 0.1 | 0.1 | - | 0.3 | - |
| Elbow | 14.2 | 14.7 | 15.2 | 15.2 | 11.0 | 6.5 | 18.4 | 20.4 |
| Wrist | 27.7 | 28.5 | 22.7 | 23.1 | 30.4 | 29.6 | 24.6 | 21.8 |
| CM | 0.5 | 0.3 | 0.8 | 0.7 | 1.2 | 1.6 | 1.1 | 1.2 |
| MCP | 14.5 | 15.2 | 21.8 | 21.6 | 5.4 | 4.3 | 2.9 | 1.8 |
| (P)IP | 12.8 | 7.8 | 11.9 | 11.5 | 2.9 | 4.3 | - | - |
| DIP | 0.1 | 0.0 | 0.3 | 0.2 | 0.1 | 0.5 | - | - |
| Hip | 0.1 | 0.4 | 0.5 | 0.4 | 2.1 | 1.9 | 1.6 | 1.2 |
| Knee | 15.3 | 16.2 | 10.9 | 10.7 | 16.5 | 17.6 | 12.2 | 11.0 |
| Ankle | 0.8 | 1.1 | 2.6 | 2.7 | 6.6 | 5.4 | 11.6 | 13.3 |
| Midfoot | - | 0.5 | 1.0 | 1.1 | 1.5 | 2.5 | 1.7 | 0.4 |
| Forefoot | 0.2 | 1.0 | 1.2 | 1.1 | 4.1 | 4.1 | 1.2 | 1.1 |

but slightly different contributions from the joints within the VAS group. HAQ-DI and mHAQ also showed similar contributions, but the wrist seemed to contribute more to HAQ-DI whereas the shoulder contributed to mHAQ; this heterogeneity should raise caution when using these two separate tools.

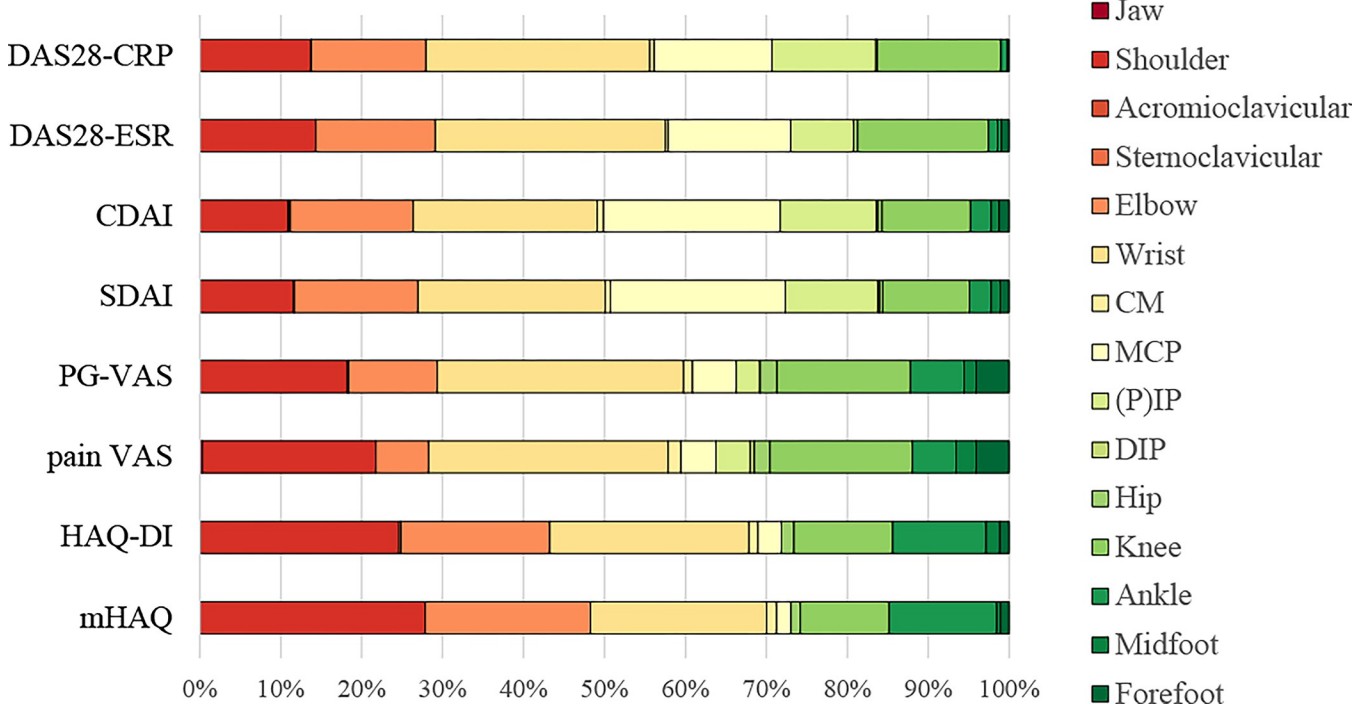

**Fig 1. Relative contribution of various joints to DAS, DAI, VAS, and HAQ in the 2013–2019 KURAMA cohort.** The contribution ratio of each joint is summarized in the graph. The contribution ratio was calculated using the ratio of the partial R-squared values as a result of multivariate analysis.

## Joint symptoms differentially contribute to pain, disease activities, and functional disabilities

From the joint perspective, the relative contributions of joint symptoms to these indexes are shown in Fig 2 and S6 Fig. The contribution of the upper extremity joints consists of most of all indexes. In particular, the wrist was the highest contributing joint in most of the indexes, especially in the VAS group. The MCP joint has an important contribution to the disease activity, especially in the DAI group, whereas the PIP joint contributed equally to these indexes. In contrast, the MCP and PIP joints contributed much less to the VAS and HAQ groups, indicating the clinical importance of these small joints in the disease activity but not in pain or daily function. The shoulder and elbow showed similar significance in contributions to all indexes except the HAQ group, in which the contribution was much more. The elbow contributed more in the DAI group than the shoulder but less in the VAS group. The shoulder showed a similar contribution to the wrist in the VAS and HAQ groups, especially in the mHAQ, indicating the crucial contribution of the joint in daily activity. Lower extremity joints did not show much contribution, except for the knee, which showed apparent and expected contributions, especially in the VAS and HAQ groups. A unique contribution of the ankle was apparent even in disease activity assessments that did not have joint counts for the ankle, and the HAQ group also demonstrated a considerable contribution from the ankle. The forefoot slightly contributed to the HAQ group, while the hip had minimum contributions for all tools. Based on these results, an example of the differences after improvement or aggravation of symptoms in each joint for each assessment tool is shown in Fig 3.

## Validation of the associations between joint symptoms and assessment index

To determine whether the associations described above correspond to other sets of data, we used the 2012 KURAMA cohort, which had worse disease activity and functional disability.

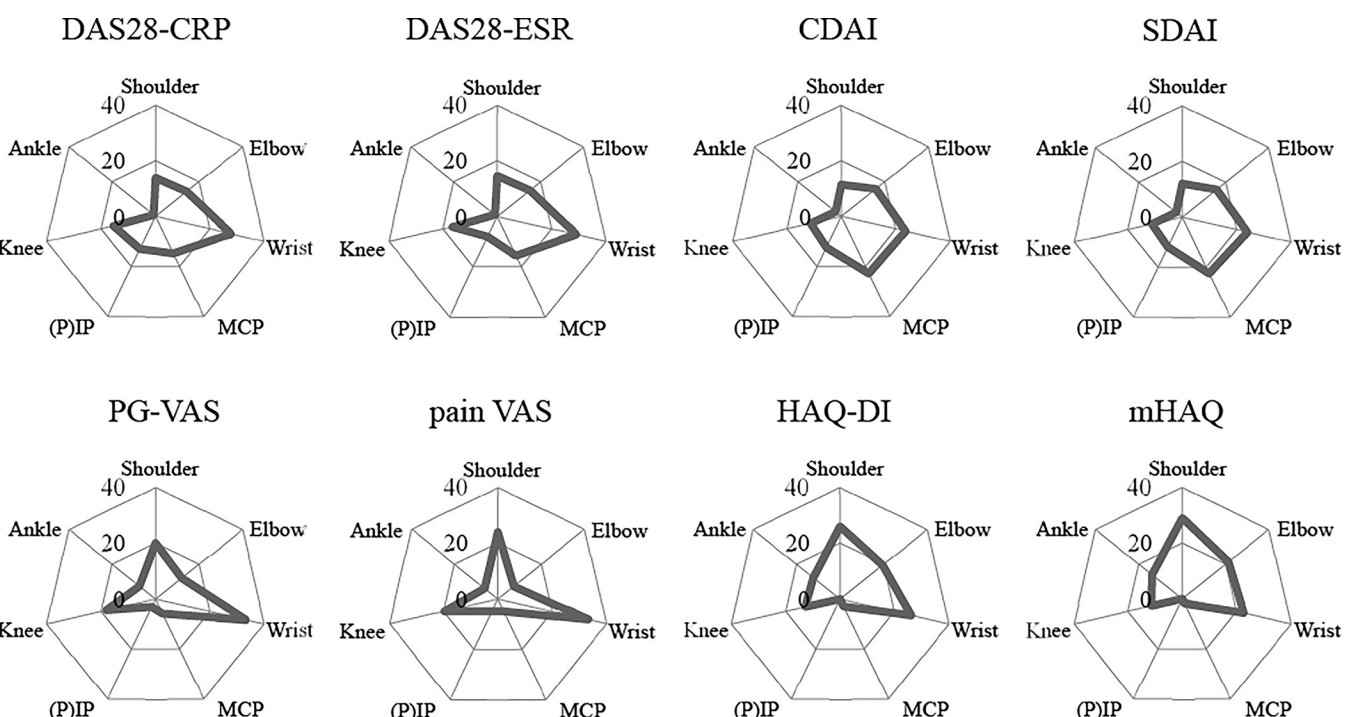

**Fig 2. Radar chart with relative contribution of various joints in the 2013–2019 KURAMA cohort.** The contribution ratio of each joint is summarized in the radar chart. Joints whose partial R-squared values were less than 0.01 were excluded.

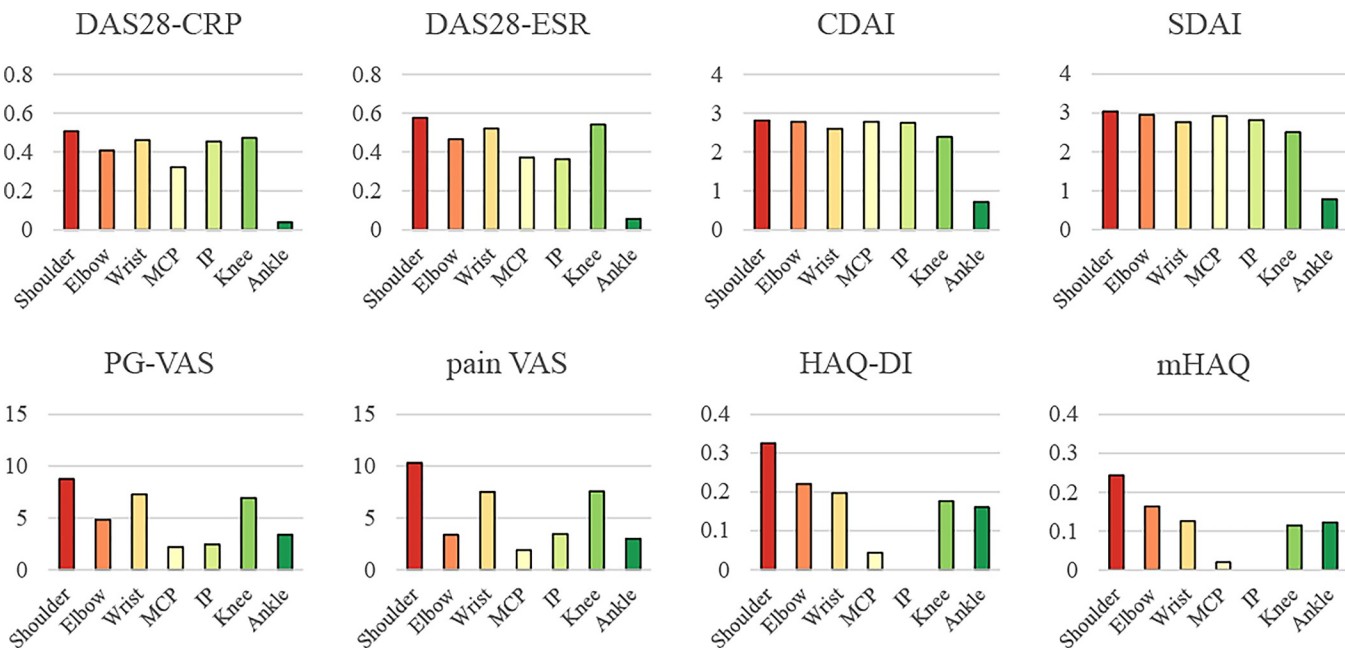

**Fig 3. Bar chart with partial regression coefficient of various joints in the 2013–2019 KURAMA cohort.** The amount of difference after symptom improvement or aggravation of each joint is summarized in a bar graph. Joints whose partial R-squared values were less than 0.01 were excluded.

S7–S10 Figs show slight differences between the main results and these results; for example, the 2012 cohort showed much more contribution of the shoulder in the VAS group and the elbow in the HAQ group than in the 2013–2019 cohort. Furthermore, the ankle joint contributed much more to the HAQ group in the 2012 cohort. However, the results obtained from both studies appear similar, indicating the validity of joint contribution to these indexes in a variety of disease statuses.

To confirm the results described above, we reanalyzed the whole data using 2012–2018 as the main analysis and 2019 as validation one. Almost the similar results were obtained from the main analysis as shown S11A Fig. In the validation study using 2019 data, the contribution of the knee and MCP joints tended to be higher overall, but the rest of the data showed approximately the similar trend (S11B Fig).

### Overall association between joint symptoms and assessment indexes

The overall association was analyzed using a Circos graph (Fig 4). As shown clearly, the joints of the upper extremity contributed more than those of the lower extremity, with the highest contributions from the wrist and shoulder. Moreover, the large joints contributed more than the small joints overall, but the MCP and PIP joints made important contributions to the DAS and DAI groups. In contrast, the ankle played a small but important role in most assessment indexes, especially in the HAQ group.

### Discussion

In this study, we analyzed the relative contribution of each joint to clinical assessment tools using 4016 patients from a large RA cohort. Multiple regression analysis for disease activity (DAS and DAI groups), global symptoms and pain (VAS group), and functional disability (HAQ group) revealed that upper extremity joints contributed more than lower extremity

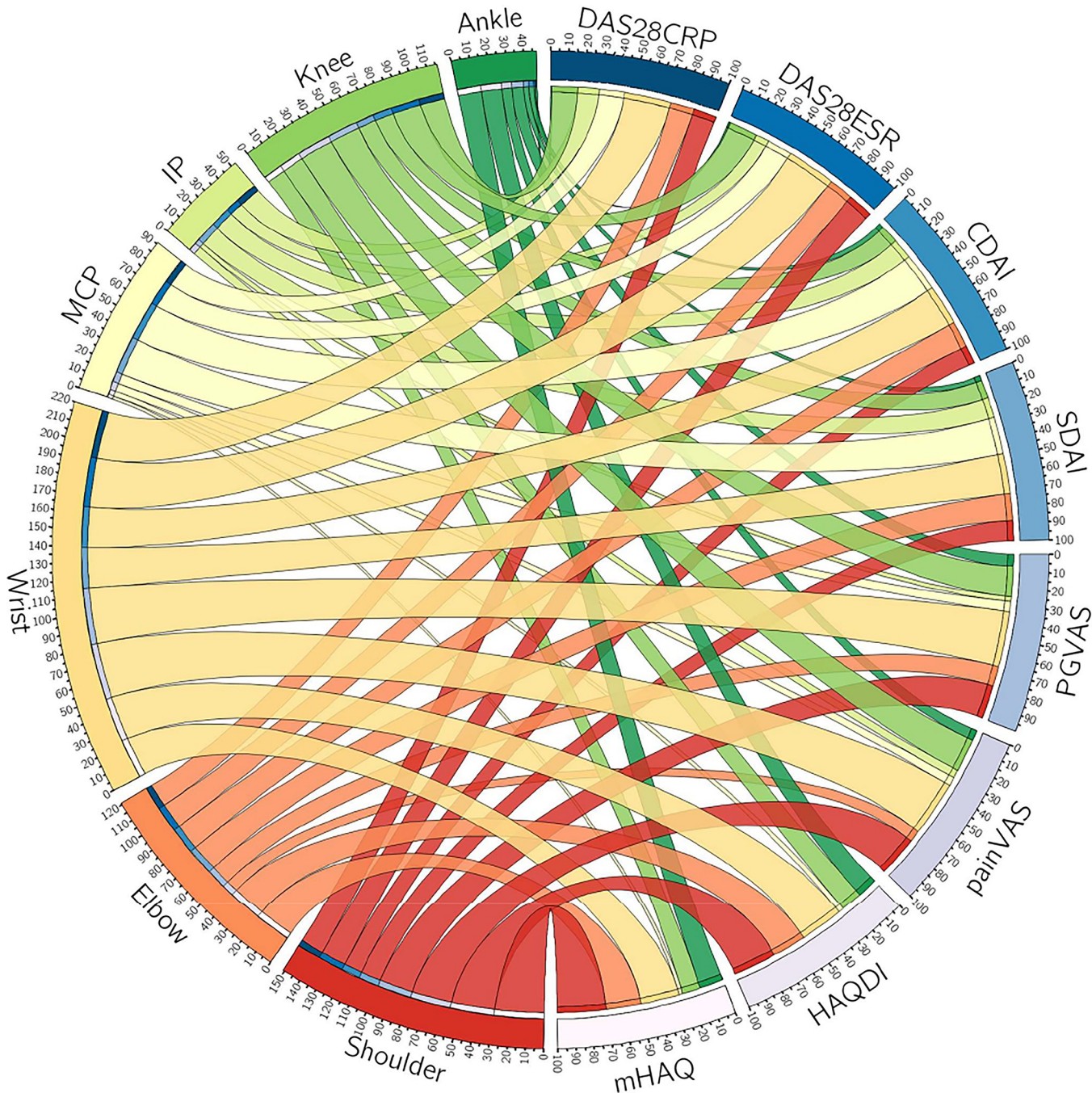

**Fig 4. Circos graph with relative contribution of various joints in the 2013–2019 KURAMA cohort.** The contribution ratio of each joint is summarized in the Circos graph. Joints whose partial R-squared values were less than 0.01 were excluded.

joints. In particular, the wrist had the highest contribution to most indicators. However, the shoulder, elbow, MCP, and PIP joints showed unique and differential contributions to these indexes. Moreover, the large joints contributed more than the small ones. When the wrist was included as a large joint, the large joints (the shoulder, elbow, wrist, hip, knee, and ankle joints) showed > 70% of all joint contributions in each index, especially in the HAQ group (> 90%). On the other hand, the hip showed negligible contribution to any of the indexes whereas the

ankle joints played a small but important role in most of the indicators evaluated, especially in the HAQ group. To the best of our knowledge, this is the first study to summarize the contribution of each joint to the frequently used clinical indexes typically applied in rheumatology practice.

The study by Tanaka et al. is one of the few studies assessing joint contributions to disease assessment tools [10]. Using the ACR Core Data Set, they examined the impact of each joint on the VAS and HAQ scores. The results showed that the shoulder, elbow, and knee joints contributed the most to the VASs and HAQs, followed by the wrist and ankle. The total contribution of the shoulder, elbow, and knee joints was approximately 70% and 90% after the addition of the wrist and ankle, respectively. Overall, they showed similar critical contributions from the large and upper extremity joints as those in our study. However, the results of our study showed much more wrist contribution in the VAS and HAQ groups. On the other hand, the elbow showed less contribution in the VAS group but more contribution in the HAQ group. Possible reasons for the differences between the two studies are unknown, but an obvious reason is that bDMARDs and tsDMARDs were rarely used at the time of their study in 2000 [16]. The analytical methods used also seem to differ between the two studies. Moreover, they did not analyze any assessment tools for disease activity, such as the DASs and DAIs. However, despite these differences, the similarities between these two studies appear to validate the results of this study. Further, the other two studies analyzing the joint contribution to HAQ also concluded the importance of large joints, supporting the assumption in our study [11, 12]. However, this notion should be assessed repeatedly under a variety of medical circumstances.

To the best of our knowledge, this is the first study to analyze the joint contribution of any disease activity index for RA. One of the difficulties in analyzing joint contribution to the disease activity indexes is that the joint symptoms themselves are included in the calculation of each score. This is an apparent reason why the shoulder, elbow, wrist, MCP, PIP, and knee seem to contribute more than other joints. Moreover, one patient had 10 MCP (MTP) and PIP (IP for thumb and big toe) joints and 8 of the DIP joints, but the other joints such as the shoulder, elbow, and knee had only 2, which is an issue in assessing the comparative contribution in each joint. However, when one cautiously considers this, the results of this study will be useful for clinical practice. For example, the MCP joints contributed more to the DAI group than to the DAS group. Therefore, DAIs tend to be higher than DASs when the patient has symptoms in the MCP joint. As another example, the MCP and PIP joints contribute less to the DAS28-ESR than to the DAS28-CRP. Hence, it is more difficult for DAS28-ESR to be higher than DAS28-CRP when these joints are symptomatic. These factors should be considered when treating a patient with the presence of symptoms in these joints. In clinical practice, the results of this study can be applied to consider which assessment tools are likely to be affected by which joints when treated and to help determine treatment strategies. Hypothetical differences are shown in Fig 3. When a patient has a strong pain complaint and is having trouble in daily living, for example, treatment of the shoulder, in particular, (e.g., intra-articular injections) may help alleviate the patient's VAS. For patients with functional impairments in daily living, prioritizing treatment of the elbow in addition to the shoulder may also decrease the patient's HAQ. Although each patient has different joints that are tender and/or swollen, considering which of the patient's items you wish to relieve will provide the basis for determining which joints should be prioritized for treatment.

On the contrary, the results in this study demonstrate little contribution from several joints, such as the hip and MTP joints (forefoot), although the importance of these joints is clinically obvious. For example, a previous study showed that hallux valgus and forefoot pain are statistically associated with DAS28 and HAQ scores [17]. Total hip joint arthroplasty has been

conducted in a number of patients with RA, although the prevalence may be lower than previously reported [18–21]. The reason for this difference is as follows: 1) the prevalence of hip joint involvement is less and much more insidious, and 2) the MTP joints are not counted in the disease activity indexes such as DAS or DAI groups and tend to be hidden from clinicians because of footwear. However, these joints are important in clinical practice, and rheumatologists and healthcare professionals should pay sufficient attention to these joints.

This study has several limitations. First, these results came from evaluating a single group of patients at a single institution during a limited period, although we included data from a variety of patients with disease activity and functional disability receiving a variety of treatments over a span of 7 years. Second, we only included the data from one time point for each patient, and no causative relationship could be concluded from the data in this study. Third, we analyzed the contribution of joint swelling and tenderness to their resulting joint symptoms. This may elucidate only one aspect of joint symptoms, and other aspects such as limited range of motion and resting or moving pain of each joint were excluded, which would contribute to the assessment tools in different ways. Finally, as described above, different analytical methods should be considered to validate the results obtained from this study.

In conclusion, we analyzed a dataset of 4016 patients from a large RA cohort to assess joint symptom contribution to assessment tools used for disease activity, global symptoms, pain, and functional disability. This study revealed that large joints and joints in the upper extremities contribute more to the assessment tools. However, each joint provides a unique contribution for each of these assessment tools used in clinical practice. The improvement or aggravation of symptoms in each joint affects the assessment tools in different manners.

## Supporting information

**S1 Fig. Flow chart of this study.**
(TIFF)

**S2 Fig. Relative contribution of various joints, including residuals, to DAS, DAI, VAS, and HAQ in the 2013–2019 KURAMA cohort.** The contribution ratio of each joint is summarized in the graph. The contribution ratio was calculated using the ratio of the partial R-squared values including residual as a result of multivariate analysis. Joints whose partial R-squared values were less than 0.01 were excluded.
(TIFF)

**S3 Fig. Radar chart showing the relative contribution of 15 joints in the KURAMA cohort from 2013 to 2019.**
(TIFF)

**S4 Fig. Bar chart showing 15 joints contribution to DAS, DAI, VAS, and HAQ in the KURAMA cohort from 2013 to 2019.**
(TIFF)

**S5 Fig. Bar chart showing contribution of DAS, DAI, VAS, and HAQ to 15 joints in the 2013–2019 KURAMA cohort.**
(TIFF)

**S6 Fig. Bar chart with relative contribution of various joints in the 2013–2019 KURAMA cohort.** The contribution ratio of each joint is summarized in the bar chart. Joints whose partial R-squared values were less than 0.01 were excluded.
(TIFF)

**S7 Fig. Relative contribution of various joints to DAS, DAI, VAS, and HAQ in the 2012 KURAMA cohort.** The contribution ratio of each joint is summarized in the graph. The contribution ratio was calculated using the ratio of the partial R-squared values as a result of multivariate analysis. Joints whose partial R-squared values were less than 0.01 were excluded.
(TIFF)

**S8 Fig. Radar chart with relative contribution of various joints in the 2012 KURAMA cohort.**
(TIFF)

**S9 Fig. Bar chart showing 15 joints contribution to DAS, DAI, VAS, and HAQ in the 2012 KURAMA cohort.**
(TIFF)

**S10 Fig. Bar chart showing contribution of DAS, DAI, VAS, and HAQ to 15 joints in the 2012 KURAMA cohort.**
(TIFF)

**S11 Fig.** Relative contribution of various joints to DAS, DAI, VAS, and HAQ in the 2012–2018 (A) and in the 2019 (B) KURAMA cohort.
(TIFF)

**S1 Table. 1.** Multivariable association for the contribution of various joint to DAS28-ESR in the 2013–2019 KURAMA cohort. **2.** Multivariable association for the contribution of various joint to CDAI in the 2013–2019 KURAMA cohort. **3.** Multivariable association for the contribution of various joint to SDAI in the 2013–2019 KURAMA cohort. **4.** Multivariable association for the contribution of various joint to pain VAS in the 2013–2019 KURAMA cohort. **5.** Multivariable association for the contribution of various joint to mHAQ in the 2013–2019 KURAMA cohort.
(ZIP)

**S1 File. Complete study data.** Data file containing all reported parameters for all patients included in the study.
(XLSX)

## Acknowledgments

The authors thank to Profs. Chikashi Terao and Takao Fujii for their thoughtful discussion. The authors also thank Wataru Yamamoto for their technical assistance.

## Ethics approval and consent to participate

This study was designed in accordance with the Helsinki Declaration and approved by the Ethics Committee of Kyoto University Graduate School and Faculty of Medicine (E1308, R0357). All participants provided written informed consent for enrollment in this cohort.

## Author Contributions

**Conceptualization:** Hiromu Ito.

**Data curation:** Hiromu Ito, Koichi Murata, Masao Tanaka, Takayuki Fujii, Akira Onishi, Hideo Onizawa, Shinichiro Ishie, Akinori Murakami, Kohei Nishitani, Kosaku Murakami, Hiroyuki Yoshitomi, Motomu Hashimoto, Akio Morinobu.

**Formal analysis:** Akio Umemoto, Masayuki Azukizawa, Hiroyuki Yoshitomi.

**Funding acquisition:** Hiromu Ito, Koichi Murata, Masao Tanaka, Motomu Hashimoto, Shuichi Matsuda.

**Investigation:** Akio Umemoto.

**Supervision:** Masao Tanaka, Akio Morinobu, Shuichi Matsuda.

**Visualization:** Masayuki Azukizawa, Hiroyuki Yoshitomi.

**Writing – original draft:** Akio Umemoto, Hiromu Ito.

**Writing – review & editing:** Akio Umemoto, Hiromu Ito, Masayuki Azukizawa, Koichi Murata, Masao Tanaka, Takayuki Fujii, Akira Onishi, Hideo Onizawa, Shinichiro Ishie, Akinori Murakami, Kohei Nishitani, Kosaku Murakami, Hiroyuki Yoshitomi, Motomu Hashimoto, Akio Morinobu, Shuichi Matsuda.

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
