## [Decision Letter · Decision Letter 0]

12 Mar 2023

PONE-D-23-02162

How do symptoms of each joint contribute to global pain, disease activity and functional disability in rheumatoid arthritis? -a comprehensive association study using a large cohort-

PLOS ONE

Dear Dr. Ito,

Thank you for submitting your manuscript to PLOS ONE. After careful consideration, we feel that it has merit but does not fully meet PLOS ONE’s publication criteria as it currently stands. Therefore, we invite you to submit a revised version of the manuscript that addresses the points raised during the review process.

We look forward to receiving your revised manuscript.

Kind regards,

Shunsuke Mori, MD, PhD

Academic Editor

PLOS ONE

Journal Requirements:

    "I have read the journal's policy and the authors of this manuscript have the following competing interests: AO, HO, TF, KMurat, and MT belong to the department financially supported by two local governments in Japan (Nagahama City, Shiga and Toyooka City, Hyogo) and five pharmaceutical companies (Mitsubishi Tanabe Pharma Co., Chugai Pharmaceutical Co., Ltd., AYUMI Pharmaceutical Co., Asahi Kasei Pharma Co. and UCB Japan Co., Ltd.). KURAMA cohort study is supported by grant from Daiichi Sankyo Co. Ltd. HI has received a research grant and/or speaker fee from Bristol-Myers Squibb. MT has received research grants and/or speaker fees from Abbvie Inc., Asahi Kasei Pharma Co., Astellas Pharma Inc., Chugai Pharmaceutical Co. Ltd., Daiichi Sankyo Co., Ltd., Eisai Co. Ltd., Eli Lilly Japan K.K., Janssen Pharmaceutical K.K., Kyowa Kirin Co. Ltd., Pfizer Inc., Taisho Pharmaceutical Co. Ltd., Mitsubishi Tanabe Pharma Co., Teijin Pharma Ltd.,and UCB Japan Co. Ltd. AO reports grants from Pfizer Inc., Bristol-Myers Squibb., Ayumi, The Health Care Science Institute, and Advantest, and personal fees from Asahi Kasei Pharma Co., Chugai Pharmaceutical Co. Ltd., Eli Lilly Japan K.K, Ono Pharmaceutical Co., Mitsubishi Tanabe Pharma, Eisai Co. Ltd., Abbvie Inc., Takeda Pharmaceutical Company Limited, and Daiichi Sankyo Co. Ltd.. MH receives grants and /or speaker fees from Abbvie Inc., Asahi Kasei Pharma Co., Astellas Pharma Inc., AYUMI Pharmaceutical Co., Bristol-Meyers, Chugai Pharmaceutical Co. Ltd, Daiichi Sankyo Co. Ltd, EA Pharma, Eisai Co. Ltd., Eli Lilly Japan K.K., Nihon Shinyaku, Novartis Pharma, and Mitsubishi Tanabe Pharma Co.. AU, MA,  SI, AMu, KN, KMurak, HY, AMo and SM declared no conflicts of interest. This study is conducted as an investigator initiate study. The sponsors were not involved in the study design; in the collection, analysis, interpretation of data; in the writing of this manuscript; or in the decision to submit the article for publication. The authors, their immediate families, and any research foundations with which they are affiliated have not received any financial payments or other benefits from any commercial entity related to the subject of this article."

Additional Editor Comments:

Our reviewers found some interests in this study, but pointed out a number of criticisms that require improvement and amendment. I ask the authors to fully respond to all comments made by reviewers in the revised version. Additionally, the authors should report p values with up to 2 significant digits and maximum 3 decimals (such as p < 0.001, p = 0.009, p = 0.025, and p = 0.27). This is a much used recommendation.

Reviewers' comments:

Reviewer's Responses to Questions

**Comments to the Author**

1. Is the manuscript technically sound, and do the data support the conclusions?

Reviewer #1: Yes

Reviewer #2: Yes

Reviewer #3: Yes

2. Has the statistical analysis been performed appropriately and rigorously? 

Reviewer #1: I Don't Know

Reviewer #2: Yes

Reviewer #3: Yes

3. Have the authors made all data underlying the findings in their manuscript fully available?

Reviewer #1: Yes

Reviewer #2: Yes

Reviewer #3: Yes

4. Is the manuscript presented in an intelligible fashion and written in standard English?

Reviewer #1: Yes

Reviewer #2: Yes

Reviewer #3: Yes

5. Review Comments to the Author

Reviewer #1: This study analyzes the joint contribution of any disease activity index for RA.

In particular, the focus on the hip and MTP joint is unique.

How this research can be applied or utilized in daily clinical practice is still being determined.

Reviewer #2: Comments to Author:

To examine the association between joint lesion location and disease status assessment tools, the authors performed a multivariate analysis of the relative contribution of symptoms at various joint sites to disease status assessment tools, VAS scores, and functional disability indices using data from a large Japanese RA cohort. The results showed that the disease state assessment tools was affected differently depending on the site of the symptomatic joint. I believe that this is an important study because of its relevance to daily clinical practice. I have some specific comments.

Major points:

1. Why is it that in Table 1 the patient background is quite different in 2012 compared to 2013-2019? It may be less valid to consider the 2013-2019 registrants as a general population. Has the method of patient enrollment changed since 2013? The authors should more clearly state why they chose 2012 as the dataset for the Validation study.

2. Conversely, the same analysis could be done with the 2012-2018 registrants as the main study and the 2019 registrants as the validation study. If the same results are obtained, the reliability of the results of this study will increase.

Reviewer #3: The authors tried to analyze the relative contribution of various joints to the disease scores in a large RA cohort. They found that large joints and joints in the upper extremities contributed more to the disease scores. These results were confirmed in the replication study.

Introduction and Discussion sections were too long.

“indices” and “indexes” were used in the manuscript. Please use one of them.

In line 269, [8] would be [10].

6. PLOS authors have the option to publish the peer review history of their article (what does this mean?). If published, this will include your full peer review and any attached files.

Reviewer #1: No

Reviewer #2: No

Reviewer #3: No

---

## [Author Response · Author response to Decision Letter 0]

29 Mar 2023

Point by point responses to the reviewers’ comments

First of all, we sincerely appreciate your careful evaluations. The answers for the specific comments were written below each comment in bold type. Revised text is highlighted by red.

Reviewer #1: 

This study analyzes the joint contribution of any disease activity index for RA.

In particular, the focus on the hip and MTP joint is unique.

We were delighted to hear that you acknowledge the importance of this manuscript.

How this research can be applied or utilized in daily clinical practice is still being determined.

Response: 

That is a very important point in this study. We consider that it is helpful for practitioners to consider which joint(s) should be prioritized for treatment for the particular patient, as described in line 308-316. We believe that the comparison of the results of Fig. 3 with the patient's painful and swollen joints would greatly help determine a treatment strategy.

Reviewer #2:

To examine the association between joint lesion location and disease status assessment tools, the authors performed a multivariate analysis of the relative contribution of symptoms at various joint sites to disease status assessment tools, VAS scores, and functional disability indices using data from a large Japanese RA cohort. The results showed that the disease state assessment tools was affected differently depending on the site of the symptomatic joint. I believe that this is an important study because of its relevance to daily clinical practice. I have some specific comments.

We deeply appreciate the reviewer’s comments which is rewarding to our commitments and efforts to this study.

Major points:

1. Why is it that in Table 1 the patient background is quite different in 2012 compared to 2013-2019? It may be less valid to consider the 2013-2019 registrants as a general population. Has the method of patient enrollment changed since 2013? The authors should more clearly state why they chose 2012 as the dataset for the Validation study.

Response: 

The difference in patient background data between 2012 and 2013-2019 can be attributed to the gradual changes in treatment over time. Items related to disease activity are improving, PSL use is decreasing while the use of bDMARD and tsDMARD has been increasing while there are no differences in age or disease duration. Also, nothing has been changed in patient enrollment methods.

2. Conversely, the same analysis could be done with the 2012-2018 registrants as the main study and the 2019 registrants as the validation study. If the same results are obtained, the reliability of the results of this study will increase.

Response: 

Thank you for your helpful advice. We reanalyzed the whole data using 2012-2018 as the main analysis and 2019 as validation one. Almost the same results were obtained for the main data. For 2019, the contribution of the knee and MCP joints tended to be higher overall, but the rest of the data showed approximately the similar trend (suppl Fig. 11). We added the descriptions in the results section (line 245-249).

Reviewer #3:

Introduction and Discussion sections were too long.

Response: 

Thank you for your suggestion, I have removed the redundant parts in the Introduction and Discussion sections.

“indices” and “indexes” were used in the manuscript. Please use one of them.

Response: 

Thanks for pointing that out, I have standardized on indexes.

In line 269, [8] would be [10].

Response: 

Thank you for pointing that out. You are correct and I have corrected it to [10].

(The end of the responses)

---

## [Decision Letter · Decision Letter 1]

18 Apr 2023

How do symptoms of each joint contribute to global pain, disease activity and functional disability in rheumatoid arthritis? -a comprehensive association study using a large cohort-

PONE-D-23-02162R1

Dear Dr. Ito,

We’re pleased to inform you that your manuscript has been judged scientifically suitable for publication and will be formally accepted for publication once it meets all outstanding technical requirements.

Kind regards,

Shunsuke Mori, MD, PhD

Academic Editor

PLOS ONE

Reviewers' comments:

Reviewer's Responses to Questions

**Comments to the Author**

1. If the authors have adequately addressed your comments raised in a previous round of review and you feel that this manuscript is now acceptable for publication, you may indicate that here to bypass the “Comments to the Author” section, enter your conflict of interest statement in the “Confidential to Editor” section, and submit your "Accept" recommendation.

Reviewer #1: All comments have been addressed

Reviewer #2: All comments have been addressed

2. Is the manuscript technically sound, and do the data support the conclusions?

Reviewer #1: Yes

Reviewer #2: Yes

3. Has the statistical analysis been performed appropriately and rigorously? 

Reviewer #1: Yes

Reviewer #2: Yes

4. Have the authors made all data underlying the findings in their manuscript fully available?

Reviewer #1: Yes

Reviewer #2: Yes

5. Is the manuscript presented in an intelligible fashion and written in standard English?

Reviewer #1: Yes

Reviewer #2: Yes

6. Review Comments to the Author

Reviewer #1: (No Response)

Reviewer #2: To examine the association between joint lesion location and disease status assessment tools, the authors performed a multivariate analysis of the relative contribution of symptoms at various joint sites to disease status assessment tools, VAS scores, and functional disability indices using data from a large Japanese RA cohort. The results showed that the disease state assessment tools were affected differently depending on the site of the symptomatic joint. I believe that this is an important study because of its relevance to daily clinical practice. The authors have well responded to the reviewers’ comments.

7. PLOS authors have the option to publish the peer review history of their article (what does this mean?). If published, this will include your full peer review and any attached files.

Reviewer #1: No

Reviewer #2: No

---

## [Editor Report · Acceptance letter]

5 May 2023

PONE-D-23-02162R1 

How do symptoms of each joint contribute to global pain, disease activity and functional disability in rheumatoid arthritis? -a comprehensive association study using a large cohort- 

Dear Dr. Ito:

I'm pleased to inform you that your manuscript has been deemed suitable for publication in PLOS ONE. Congratulations! Your manuscript is now with our production department. 

Kind regards, 

on behalf of

Dr. Shunsuke Mori 

Academic Editor

PLOS ONE